# ReStainGAN: Leveraging IHC to IF Stain Domain Translation for *in-silico* Data Generation

**Dominik Winter**[1]                    DOMINIK.WINTER@ASTRAZENECA.COM

**Nicolas Triltsch**[1]                    NICOLAS.TRILTSCH@ASTRAZENECA.COM

**Philipp Plewa**[1]                    PHILIPP.PLEWA@ASTRAZENECA.COM

**Marco Rosati**[1]                    MARCO.ROSATI@ASTRAZENECA.COM

**Thomas Padel**[1]                    THOMAS.PADEL@ASTRAZENECA.COM

**Ross Hill**[2]                    ROSS.HILL@ASTRAZENECA.COM

**Markus Schick**[1]                    MARKUS.SCHICK@ASTRAZENECA.COM

**Nicolas Brieu**[1]                    NICOLAS.BRIEU@ASTRAZENECA.COM

[1] *AstraZeneca Computational Pathology GmbH, Bernhard-Wicki-Str. 5. 80636 Munich, Germany*

[2] *AstraZeneca, Discovery Centre, 1 Francis Crick Avenue, Cambridge, CB2 0AA, United Kingdom*

## Abstract

The creation of *in-silico* datasets can expand the utility of existing annotations to new domains with different staining patterns in computational pathology. As such, it has the potential to significantly lower the cost associated with building large and pixel precise datasets needed to train supervised deep learning models. We propose a novel approach for the generation of *in-silico* immunohistochemistry (IHC) images by disentangling morphology specific IHC stains into separate image channels in immunofluorescence (IF) images. The proposed approach qualitatively and quantitatively outperforms baseline methods as proven by training nucleus segmentation models on the created *in-silico* datasets.

**Keywords:** Generative Adversarial Networks, *in-silico* Data Generation, Computational Pathology

## 1. Introduction

Training deep learning models using *in-silico* data is a common approach to minimize the costly effort of creating large and pixel-precise labeled datasets. Tissue morphologies in stained whole slide images (WSI) are often similar but the high diversity of stains poses a considerable challenge for model generalization. Using domain translation methods, existing annotations with pixel-precise mappings can be translated to new domains, boosting efficiency of model training in computational pathology. Numerous generative methods for domain translation exist, such as CycleGANs (Jose et al., 2021) and diffusion models (Kazerouni et al., 2022; Guo et al., 2024). CycleGANs, as opposed to diffusion models are computationally efficient and can be trained on unpaired images. Applications in computational pathology range from stain normalization (Shaban et al., 2019) to transfer learning (Brieu et al., 2019, 2022) and data augmentation (Wagner et al., 2021). This work introduces ReStainGAN, a CycleGAN based approach that enables generation of new IHC images using an auxiliary immunofluorescence (IF) domain. We present an extention of existing IHC to IF domain transfer methods (Brieu et al., 2022) and a new generative approach. ReStainGAN can disentangle stain components in the IHC domain to seperate

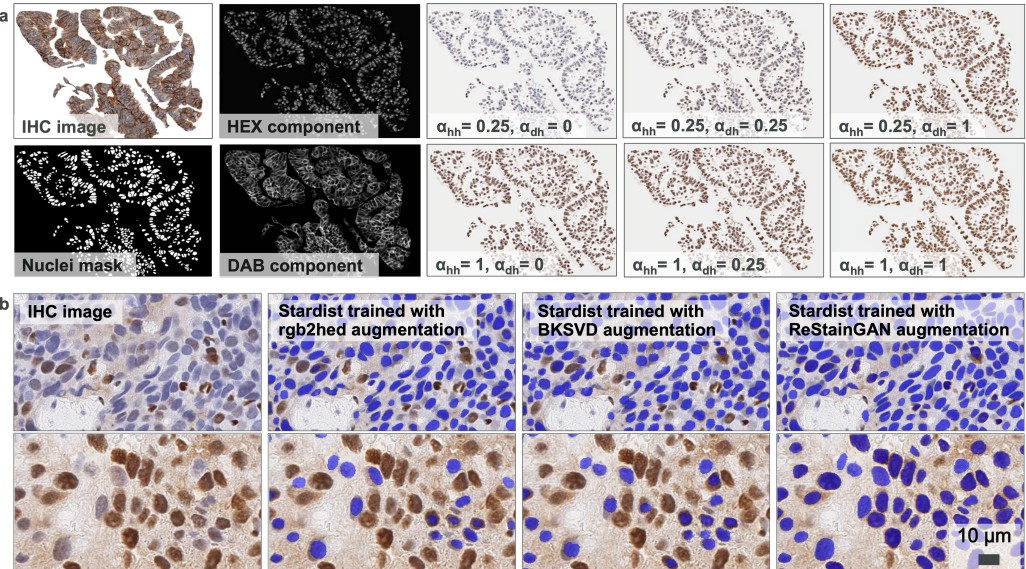

Figure 1: a) ReStainGAN disentangles nuclear and cell membrane representations in IHC cell membrane marker images through domain transfer to an auxiliary IF domain. Manipulating these DAB and HEX components with $\alpha_{hh}$, $\alpha_{dh}$ while setting $[\alpha_{hd}, \alpha_{dd}] = 0$ yields multiple versions of *in-silico* IHC nuclear marker stained images after backtranslation from the IF to the IHC domain. b) The StarDist nucleus segmentation model (blue overlay) performs best trained on *in-silico* data created with the proposed ReStainGAN (right).

channels in the IF domain. Using simple mathematical operations the IF channels can be manipulated before back-translation, generating new *in-silico* IHC images with pixel-precise preservation of morphological structures. Proving the utility of our approach we restain *in-silico* nuclear marker images from cell membrane marker images. Using existing labels of the cell membrane marker domain we train StarDist nucleus segmentation models (Weigert et al., 2020) on *in-silico* nuclear marker images without requiering addtional annotations.

## 2. Methods

Lets denote domain $A$ as a set of labeled IHC images $(x_A)_{x_A \in A}$ with DAB cell membrane marker (e.g. HER2) and hematoxylin (HEX) nuclear counter stain, and domain $B$ an unpaired set of unlabeled IF stained images $(x_B)_{x_B \in B}$ with a DAPI nuclear marker channel and a cell membrane marker channel (e.g. HER2). Because of the equivalence between the IF and the HEX-DAB (HD) domains, a bijective mapping between the RGB and HD colorspaces can be learned by ReStainGAN using two generators $\mathcal{G}_{AB}$ and $\mathcal{G}_{BA}$ performing IHC to IF and IF to IHC translation. Given a sample $x_A$ from the source domain $A$, the associated HD components $x_{HD} = \mathcal{G}_{AB}(x_A) := (x_{|H}, x_{|D})$ can be modified by the restaining function $\kappa$ and transformed back to domain $A$, yielding transformed IHC images:

$$x'_A = \mathcal{G}_{BA} \circ \kappa \circ \mathcal{G}_{AB}(x_A). \tag{1}$$

| Model performance | F1 score | Sensitivity | Precision |
|---|---|---|---|
| No augmentation | 0.604 | 0.450 | **0.920** |
| rgb2hed | 0.629 | 0.498 | 0.853 |
| BKSVD | 0.697 | 0.588 | 0.857 |
| Ours | **0.848** | **0.840** | 0.856 |

Table 1: Quantitative results for cell center detection between manual annotations and centers of the predicted cell segmentation masks of the StarDist models. Centers are matched using the Hungarian algorithm with a maximum distance of $1.5\mu m$.

In the case of restaining IHC images with cell membrane marker to IHC images with nuclear marker, the restaining function can be formulated as:

$$\kappa_\alpha(x_{HD})_{|H} = \min(\max(\alpha_{hh}x_{|H} + \alpha_{dh}x_{|D}, 0), 1)$$
$$\kappa_\alpha(x_{HD})_{|D} = \min(\max(\alpha_{hd}x_{|H} + \alpha_{dd}x_{|D}, 0), 1),$$
(2)

Modifying the parameters $(\alpha_{ij})_{i\in[h,d],j\in[h,d]}$ allows for manipulation of morphology specific stains in the IHC domain. Selecting $\alpha_{hh}$, $\alpha_{dh} > 0$ and $[\alpha_{hd}, \alpha_{dd}] = 0$ yields *in-silico* monoplex IHC images with nuclear staining of difference strength (cf. Fig. 1 a) while removing the cell membrane staining. This enables the generation of an *in-silico* dataset of IHC images stained with a nuclear marker from a labeled cell membrane marker IHC dataset.

## 3. Results

While the proposed ReStainGAN allows for creation of infinite amounts of *in-silico* data, we restricted ourselves to the six combinations defined by $\alpha_{dh} \in [0, 0.25, 1]$ for DAB and $\alpha_{hh} \in [0.25, 1]$ for HEX expression. A total of 421 training and 179 validation Field of Views (FOV) (20x - $0.5\mu m/px$) were selected on WSIs stained with a cell membrane marker (e.g. HER2) in which all cell centers were labeled by pathologists. Based on these, 2526 training and 1074 validation *in-silico* FOVs were generated using ReStainGAN. As baselines we employed the original FOVs, rgb2hed function of scikit-image (Van der Walt et al., 2014) and Bayesian K-SVD (BKSVD) (Pérez-Bueno et al., 2022) transformation for Hematoxylin and DAB color channel seperation analogous to ReStainGANs color seperation. In total, four StarDist models were trained using these datasets and were evaluated on the same 49 FOVs in 27 test-set WSIs stained with a nuclear marker (e.g. Ki67) and a total of 14.987 cell centers manually annotated by pathologists. StarDist models trained with data generated using ReStainGAN outperforms the baseline methods (see Fig. 1 b and Tab. 1).

## 4. Discussion

We propose ReStainGAN, a generative model that leverages auxilary IF domains for disentangling stain components in IHC images and thereby enables the generation of various different nuclear marker *in-silico* images. Application to the downstream task of nucleus segmentation demonstrates the superiority of the method as compared to baseline methods. It has already been shown that this method can be generalized from monoplex to duplex IHC assays (Brieu et al., 2024). Future work includes application to other downstream tasks, such as the semantic segmentation of epithelium regions.

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
