# OpenReview forum: "ReStainGAN: Leveraging IHC to IF Stain Domain Translation for in-silico Data Generation"
_MIDL.io/2024/Short_Papers — MIDL 2024 Short Papers_

### Official Review · Reviewer_Ws23 · 2024-04-16

**Confidence:** 5
**Final Rating:** 4

**Review:**

The paper presents ReStainGAN, an generative model that separates stain components in IHC images using auxiliary IF domains. This method enables the generation of in-silico IHC images, marking a methodological progression in the field. The model's efficacy is demonstrated in nucleus segmentation, where it outperforms established baseline methods. The potential for applying ReStainGAN to other tasks, such as semantic segmentation of epithelial regions, underscores its versatility and scalability.

However, the paper's primary limitation is its lack of detailed explanation about how ReStainGAN differentiates from existing GAN techniques. Additionally, there is uncertainty regarding the model's generalizability across different centers, samples, and scanners, raising questions about its broad applicability in diverse clinical settings.

---

### Decision · Program_Chairs · 2024-04-26

Accept